# "Cold-Dry" and "Cold-Wet" Events in the Late Holocene, Southern Russian Far East

**Nadezhda Razjigaeva** [1],*, **Larisa Ganzey** [1], **Tatiana Grebennikova** [1] and **Vladimir Ponomarev** [2]

1   Pacific Geographical Institute FEB RAS, Radio St., 7, 690041 Vladivostok, Russia
2   V.I. Il'ichev Pacific Oceanological Institute FEB RAS, Baltiyskaya, 43, 690041 Vladivostok, Russia
*   Correspondence: nadyar@tigdvo.ru; Tel.: +7-924-237-89-81

**Abstract:** Two late Holocene cold events were described for the Southern Russian Far East: 2800–2600 year BP and the Little Ice Age (LIA) (~1300–1850 CE). The synthesis is based on multi-proxy records on profile "the mainland (Primorye)-islands (Sakhalin-Kurils)". Main archives are sediments of small lakes and peat bogs that recorded the high-resolution environmental changes. The temporal resolution of reconstructions here is up to 26–40 years. During the cold event of 2800–2600 year BP, the humidity decreased sharply, there were long-term dry seasons without strong floods, and buried soil formed on floodplains. The LIA was wet in the mainland and the Kurils, with frequent strong floods, but was drier in Sakhalin. The cooling was characterized by sharp temperature fluctuations where high moisture conditions alternated with short-term drier periods. The shift in geographical position and intensity of the main centers of atmospheric action caused a paleoclimatic interpretation of these events (Aleutian Low, Siberian and North Pacific Highs, the summer Okhotsk anticyclone and the Far East depression). Changes in the North Pacific oscillations played an important role in the alternation of cold-dry and cold-wet periods. Anomalies in the intensity of El Niño and the monsoon system led to changes in tropical and extratropical cyclone trajectories and cyclogenesis in general.

**Keywords:** cold event 2800–2600 year BP; Little Ice Age; Siberian High; Aleutian Low; monsoon; cyclogenesis; ENSO; floods and droughts

## 1. Introduction

Cold events of the Holocene, consistent with the Bond events [1,2], are global in nature and have been recorded in many regions of the world [3]. It is assumed that the most important forcing mechanism of cooling was a decrease in total solar irradiance [2,4–8]. The intensity and duration of cold events varied in different regions due to the large-scale features of atmospheric circulation, the interaction of air masses at land–sea boundaries and processes in the ocean–atmosphere system, as well as such local factors as sea currents and relief that largely determine microclimatic variability, etc. Three global cold events were identified for the late Holocene [3], which manifested clearly in East Asia: 2800–2600 year BP (Neoglacial or Bond event 2), 1650–1450 year BP and 650–450 year BP (beginning of the Little Ice Age, Bond event 1). As a rule, in the temperate latitudes of the region, they were accompanied by aridization, but some records indicate an increase in precipitation [9–11].

Both in the instrumental period and on the Holocene paleo-scale, the alternation of weakening/strengthening of the meridional/zonal circulation in the atmosphere and its centers of action accompanies the weakening/strengthening of the Asian Pacific Monsoon System [12–16]. It is associated with negative/positive anomalies in tropical and extratropical cyclone activity and precipitation over the East Asian and North Western Pacific marginal zone. Changes in the North Pacific oscillations such as the El Niño/Southern Oscillation (ENSO) [15,17], the Pacific decadal oscillation (PDO) [18,19] and the North Atlantic oscillation [16,20,21] play an important role in the alternation of cold-dry and cold-wet periods during middle–late Holocene in East Asia, including China, Korea, and Japan.

The data presented by Wanner with co-authors [3] show that not all natural archives responded to the short-term cold events of the Holocene. The study of hydroclimatic changes over the past millennia in East Asia, based on the synthesis data for large areas, show significant regional variations in moisture/precipitation changes and often moisture anomalies of opposite signs in different regions [3,12]. Based on extensive evidence, it has been established that the main mechanism of climate shifts in the region is not only solar variability, but also a complex interaction of processes in the land–ocean–atmosphere [9,11,13,22–27].

Studies synthesizing paleoclimatic records [3,4,28] provide no information on the Russian Far East. To date, countless reconstruction processes on the paleoenvironment with high temporal resolution and the investigation of hydroclimatic changes have been performed for the region [29–35]. For the last 400 years, there are records based on tree rings and temperature estimates on geothermal data [36–39]. Using the example of two late Holocene cold events contrasting in moisture, we would like to consider the factors driving hydroclimatic changes in the Southern Russian Far East. The purpose of this article is to analyze the manifestation of cooling during 2800–2600 year BP and the Little Ice Age (LIA) on the mainland and the islands, and to provide a paleoclimatic interpretation of cold events with opposite moisture anomalies. It is important to know how widespread Bond events are in Northeast Asia (data for this area are plentiful, but little available) in order to determine general patterns in climate change in the Pacific and the Atlantic. An important objective of the synthesis paleoclimatic data for late Holocene cold events with different moisture trends is to better understand how atmospheric anomalies and North Pacific Ocean dynamics can contribute to a better understanding of the mechanisms of climate change in the past. This knowledge is also useful for the modern variability of the climate system to better identify link ages between its components and define forcing in widespread changes in the future.

## 2. Present Climate of Study Area

The area under research is located at the northern limit of the East Asian Monsoon and is characterized by uneven moistening by seasons; up to 80% of annual precipitation occurs in the summer [40]. The annual amount of precipitation in the region varies from 551 to 884 mm on the mainland, with the minimum observed near the Khanka Lake, and higher values in the north of the region and near the Japan Sea coast (www.primgidromet.ru, (accessed on 2 October 2022)). The amount of precipitation varies greatly, e.g., in Vladivostok, annual precipitation was 510 mm in 1997 CE and reached 1275 mm in 1974 CE [41]. The annual precipitation on the islands varies from 753 to 990 mm in Southern Sakhalin and reaches up to 1255 mm in the Southern and Central Kurils [42]. Cyclonic activity is strongly influenced by the intensity and position of the atmospheric centers of action. Atmospheric circulation in winter is determined by the interaction of the Siberian High (SH) and the Aleutian Low (AL) (Figure 1). Moreover, the position of the center mode and the intensity of SH largely determines the development of cyclonic activity not only in the winter, but also in the spring–summer season [43,44]. The North Pacific (Hawaiian) High (NPH) weakens in the winter season [45]. Dry air masses dominate over the continent throughout most of the winter (winter monsoon); snowfalls are associated with the arrival of predominantly southern cyclones originating in the Yellow, East China, and Japan Seas, as well as in the Pacific Ocean east of Japan [40,46]. The frontal zone forming here, including the continental margin, is one of the most intense in the Northern Hemisphere; cyclones move along different trajectories depending on its structure and evolution, as well as primary orientation [47].

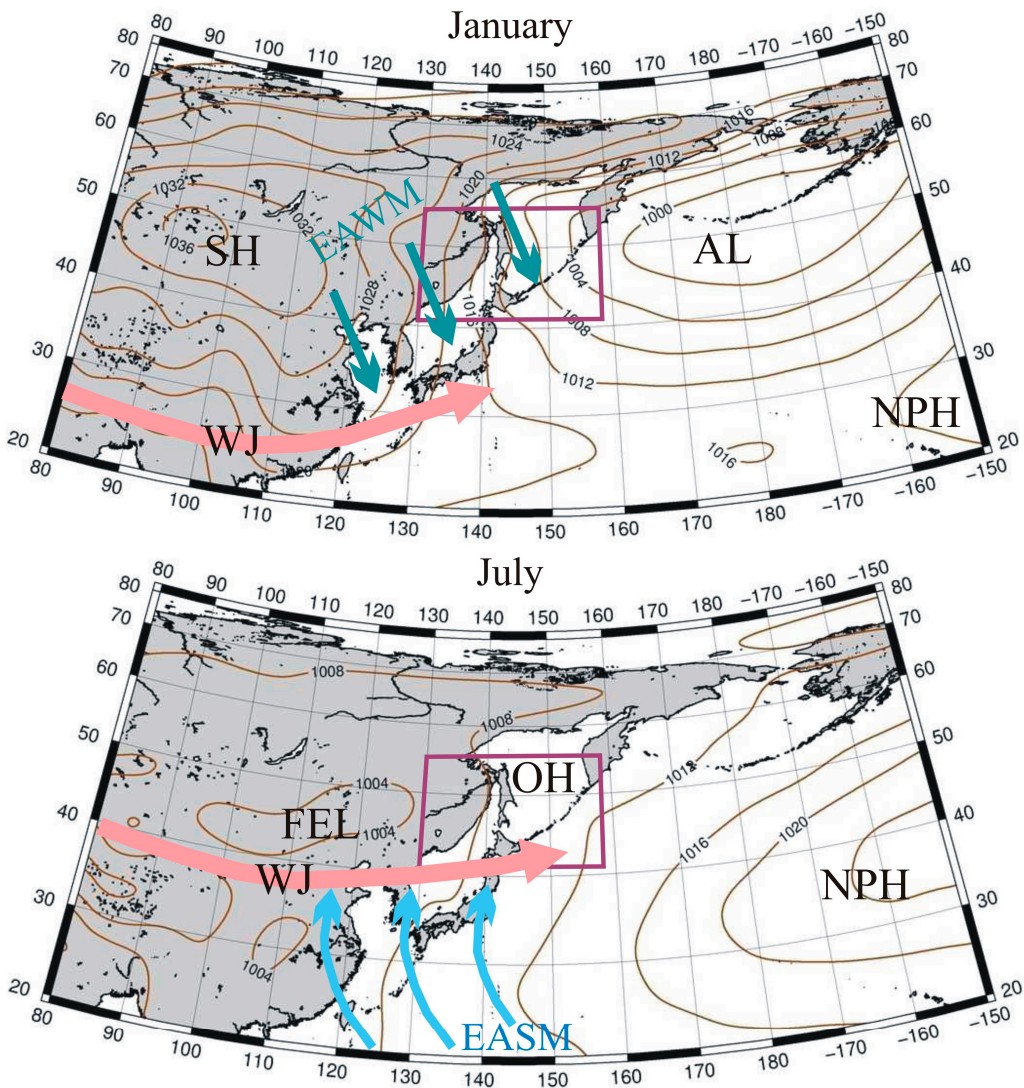

**Figure 1.** Atmospheric circulation patterns and the position of atmospheric centers of action in East Asia and the North Pacific in terms of surface pressure in January and July averaged from 1981 CE to 2010 CE (http://ferhri.org, (accessed on 27 March 2023)) with study area. EAWM and EASM stands for "East Asian winter monsoon" and "East Asian summer monsoon", WJ—"Westerly Jet".

On the islands, the situation is different. Dry masses passing over the sea get saturated with moisture and snowfalls are typical here all winter, even though heavy snowfalls are still associated with the arrival of southern and southwestern cyclones. The main amount of precipitation falls in summer–early autumn when the transfer of air masses from the ocean intensifies (summer monsoon). In spring, the breakup of SH into separate centers leads to the formation of the Okhotsk High (OH), while the Far East Low (FEL) forms over the continent, its state associated with the position and intensity of SH [44,48]. The OH is unstable, blocks the penetration of cyclones from the west, and contributes to the development of damp cold weather [45]. Atmospheric circulation is also determined by the action of NPH. An unstable high-pressure area that forms over the Japan Sea sometimes merges with its crest. Particularly extreme precipitation events are associated with the arrival of tropical cyclones (typhoons) whose trajectories are largely determined by the position of the western ridge of NPH [45]. Extratropical cyclogenesis is also active, and cyclones coming along different trajectories to the mainland bring abundant precipitation. Trajectories of the cyclones depend significantly on atmospheric pressure gradients in different seasons [40].

## 3. Materials and Methods

The main natural archives for reconstructions are lacustrine sequences and peat bogs which make it possible to obtain continuous high-resolution paleoclimatic records. Synthesis of data for 43 sections was made (Table S1). The study areas on the mainland (Primorye) included the Sikhote-Alin mountains, where sections were studied at 445–900 m a.s.l. (Solontsovskie lakes–Izyubrinye Solontsi Lake, 750 m a.s.l., Nizhnee Lake, 565 m a.s.l.; paleolakes of Sergeev and Shkotovskoe plateaus, 884 and 734 m a.s.l.; Alekseevskoe Lake on Olkhovaya Mountain, 1600 m a.s.l.; Muta peat bog on the main watershed, 568 m a.s.l.; paleolake of Milogradovka (Vanchin) River Basin (445 m a.s.l.), and paleolake on the Shufan Plateau, East Manchurian Mountains (320 m a.s.l.) (Figure 2). The criteria applied to recorded environmental changes were: (1) time range adequacy to the cold events which correlated to Bond event 1 and 2; (2) chronological control using radiocarbon and tephrostratigraphy; (3) location within different landscape belts; and (4) using different types of archives, including the sites with sharp changes of sedimentary environments.

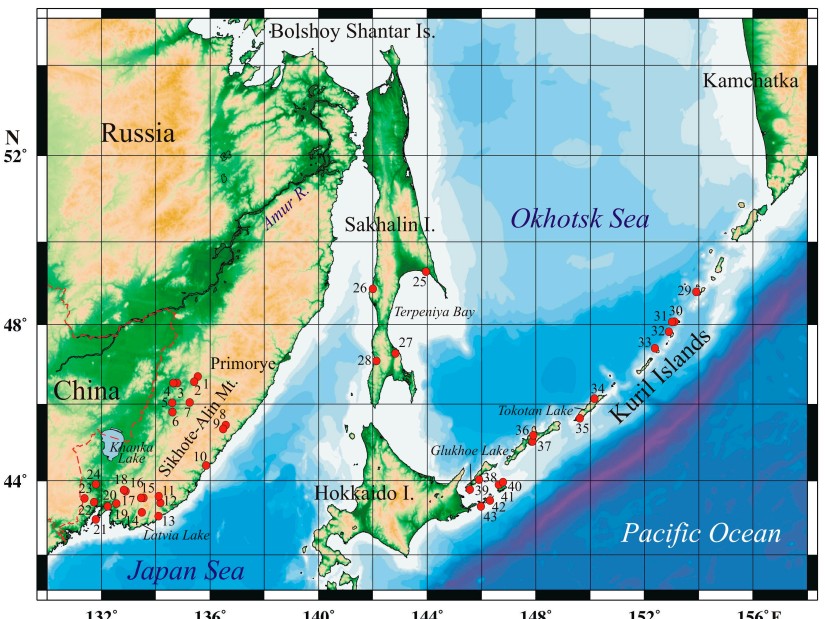

**Figure 2.** Study area and position of the sites recorded the late Holocene cold events. Continental area of South Far East: 1—Dillalakchi peat bog, Bikin River, 2—Krasny Yar peat bog, Bikin River, 3, 4—Sakhalin Mire, Bikin River, 5—Meteoritny peat bog, 6—Zveriny peat bog, 7—Glubinnoe peat bog, Bolshaya Ussurka (Iman) River Basin; 8—Izyubrinye Solontsi Lake, Solontsovskie lakes, 9—Nizhnee Lake, Solontsovskie lakes; 10—Langou I Bay, 11—Muta peat bog, 12—paleolake, Ugolniy Creek, Milogradovka (Vanchin) River Basin, 13—lagoon terrace, Kit Bay, 14—peat bog near Alekseevskoe Lake, Olkhovaya Mt., 15, 16—paleolakes, Sergeev Plateau, 17, 18—paleolake, Larchenkovo swamp, Shkotovskoe Plateau, 19—fluvial terrace, Steklyanukha River, Shkotoska River Basin, 20—Cherepash'e Lake, Murav'ev-Amursky Peninsula, 21—paleolake, Krasnaya Bay, Russian Island, 22—paleolake near Utinoe Lake, Amursky Bay coast; 23—paleolake, Shufan Plateau, 24—Starorechenskoe Fortress, Razdolnaya (Suifun) River. Sakhalin Island: 25—peat bog near Vladimirovo Settlement, 26—Orokess Terrace, Izilmetieva Bay, 27—peat bog, Naiba River Basin, 28—peat bog, Yablochnaya River Basin. Kuril Islands: 29—peat bog, Shiashkotan Island, 30, 31—soil-pyroclastic sequences and peat bog, Matua Island, 32—peat bog, ancient caldera, Rasshua Island, 33—peat bog, Ketoi Island, 34—dune field, Novokurilskaya Bay, Urup Island, 35—paleolake, Osma Bay, Urup Island; 36—Lebedinoe Lake, Iturup Island, 37—paleolake on plateau near Gniloe Lake, Iturup Island, 38—peat bog, South Kurile Ithmus, Kunashir Island; 39—paleolake near Golovnin Volcano, Kunashir Island, 40—paleolake, Khromov Bay, Shikotan Island, 41—peat bog, Gorobets River Basin, Shikotan Island, 42—paleolake, Zelenyi Island, 43—peat bog, Tanfiliev Island.

The most detailed reconstructions of natural environmental changes during the LIA were made for the Solontsovskie lakes, which were formed as a result of large landslides on the slopes of the ancient Solontsovsky Volcano in the late Holocene. Accumulation rates of sediments here reached 1.7–1.9 mm/year in the last millennium, making it possible to obtain data with a time resolution up to 26–40 years [49]. High peat accumulation rates of 0.67–0.71 mm/year were recorded on the Shufan Plateau mire in the last 1075 years and in the last 240 years–1.1–1.25 mm/year, the time resolution of reconstructions is 75–40 years [49]. The study of sections of floodplain deposit sections in the basins of the Razdolnaya (Suifun), Shkotovka, Bolshaya Ussurka (Iman), and Bikin rivers and on the sea coast was also carried out. Data were also obtained for Sakhalin Island and the Kuril Islands. On Sakhalin Island, the detailed record of strong floods was restored based on the study of a watershed peat bog with numerous loam layers [50] and the data obtained from peat bog sections by other researchers [30,51,52]. In the Kurils, the sections include tephra layers and paleotsunami deposits on the coast.

For all regions, the climatic reconstructions were made on the basis of facial and biostratigraphic studies (diatom, botanical, pollen analyses) performed according to standard methods [53–56]. The methods were described in detail in [33]. Pollen data are the main proxy for cooling recorded by the relative abundance of dark-coniferous and Korean pine pollen, broadleaved trees pollen decrease, and the presence of shrub birch pollen. Dwarf pine (*Pinus* s/g *Haploxylon*) pollen is informative for understanding winter cyclone activity since stable and deep snow cover is required. Distribution of wind-blown pollen from plants not growing on the study area indicates cyclone activity in the beginning of the vegetative period. An abundance of *Sphagnum* moss remains in peat is a good indicator of cooling. Increase in arctoboreal diatoms content show colder environments. Diatom assemblages and the botanical composition of peat were studied for the purpose of tracing changes in humidity. High concentration of valves and an increasing content of planktonic and temporary planktonic species and hydrophilous species are evidence of more humid environments. High proportions of soil diatoms indicate that the lake shallowed and the water supply to the peatlands was scant. It should be taken into account that the sections have different temporal resolutions, chronological controls, and sensitivity to climate changes.

The age of the events was determined on the basis of the radiocarbon dating of peat and paleosol carried out at the Institute of Earth Sciences, St. Petersburg State University, St. Petersburg, Russia. The age models were performed in the Bacon 2.2 program [57]. Radiocarbon dates were calibrated by the OxCal 4.4.1 program using the IntCal 20 calibration curve [58]. In the text of the article, age is given in calendar values. Tephrostratigraphy data were also taken into account. In Southern Primorye, tephra B-Tm from the caldera-forming eruption of Baitoushan Volcano (946/947 CE) was found; this tephra layer marks the last millennium. In the Southern Kurils, tephra sources were both local volcanoes and Hokkaido Island volcanoes, while in the Central Kurils they were only local volcanoes. Identification of the sources is based on the chemical analysis of volcanic glass, performed using micro-analysis by X-ray spectrometry with a LEO SUPRA 50 VP (Carl Zeiss, Oberkochen, Germany) with an energy-dispersive analyzer X-MAX 80 (Oxford Inst., Oxford, UK) in the V.G. Khlopin Radium Institute, St. Petersburg, Russia.

## 4. Paleoclimatic Records

### 4.1. Manifestation of Cooling 2800–2600 Year BP (Bond Event 2)

#### 4.1.1. Mainland

Cooling ~2630 BP caused the expansion of dark coniferous forests in the Southern Sikhote-Alin, which replaced coniferous-deciduous forests at an altitude of about 600–700 m [59]. Cold conditions are indicated by an increase in arctoboreal diatoms proportion (up to 27%) in the paleolake sediments on the Shkotovskoe Plateau, a sharp decrease in the content of planktonic diatoms (Figure 3), the paleolake turning into a swamp, and the development of species typical for soils (*Hantzschia amphioxys*). The content of arctoboreal

diatoms (*Eunotia serra*, *Pinnularia rhombarea*) reaches 58% in the Sergeev Plateau paleolake sediments (elevation 884 m a.s.l.). According to the diatom data and the botanical composition of peat, the role of atmospheric supply in the water balance of bogs increased, the level of bog waters decreased, and the mire passed into the stage of raised bogs. During cooling, the peatland plant *Scheuchzeria palustris* appeared in the swamp around western paleolake ~3010–2870 years ago and disappeared during watering decrease, while in the near eastern paleolake, *Scheuchzeria* grew 2750–2320 years ago. At the peak of cooling, sphagnum mosses became common [60].

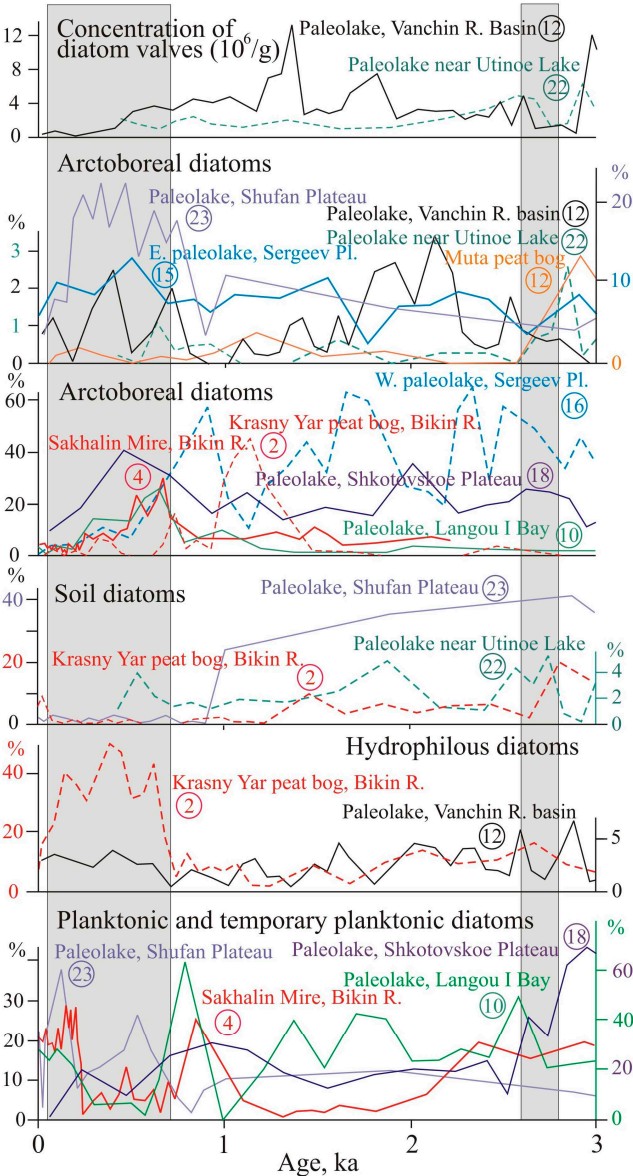

**Figure 3.** Comparison of results for diatom proxies from peat bog and paleolake sequences on the mainland. Grey bars indicate 2800–2600 year BP cold event and the LIA. Number of sites according to Figure 2.

Moisture supply reduced 3080–2735 year BP on the swampy watershed (Muta peat bog). The increasing proportion of *Pinnularia borealis* and *Hantzschia amphioxys* among diatoms suggests occasional drought periods. Species brought by floods decreased. Further desiccation of the swamp took place in 2735–2040 year BP, as suggested by a low concentration of diatoms frustules. The swamp vegetation also changed: the coldest and driest conditions were ~2620–2215 year BP, marked by increasing *Betula ovalifolia* and Ericaceae.

The swamp began to be overgrown with larch and *Sphagnum*. Increasing *Duschekia* and *Pinus* s/g *Haploxylon* pollen indicates that the area of shrub alder and shrub pine expanded at the time [61]. Watering of the swamp in the upper Milogradovka (Vanchin) River (445 m a.s.l.) decreased about 2760–2560 years ago, which led to the development of woody vegetation. Swamp waters became more acidic and saturated with humic acids. Traces of floods associated with increased cyclogenesis were recorded in 2920–2760 year BP, and regular floods began to occur during 2560–1860 year BP [62]. In Central Sikhote-Alin, the passage of frequent fires reflected drier conditions. Near Izyubrinye Solontsi Lake (750 m a.s.l.), the degree of peat decomposition sharply decreased under cooling [49]. In the mountains of the southernmost Primorye (paleolake on the Shufan Plateau) climatic conditions became drier about 3740 years ago and were especially dry from 3050 year BP up to ~1075 year BP. Frequent pyrogenic successions in swamp vegetation occurred as a result of numerous fires [33].

In the foothills of the Sikhote-Alin western macroslope (the Bikin River basin), the appearance of *Calamagrostis* in the bog vegetation signaled moisture decrease [63]. The mires were being overgrown by larch with birches. Korean pine became more common on the mountain slopes under cooling, there may have been pure Korean pine forests in the territory, and the role of broadleaf trees in the forests decreased (Figure 4). Southward, in the Bolshaya Ussurka (Iman) River valley, a long dry period began 2900 years ago [35]. Under low moisture conditions, the tree layers of larch and birch became denser in swamps. There were frequent fires there, especially strong ones ~2670–2010 years ago. Pyrogenic changes resulted in the widespread of *Ledum hypoleucum*, cotton grass, and green mosses *Polytrichum strictum*, *P. commune*, and *Aulacomnium palustre* [64]. Regular floods started from 2670 years ago.

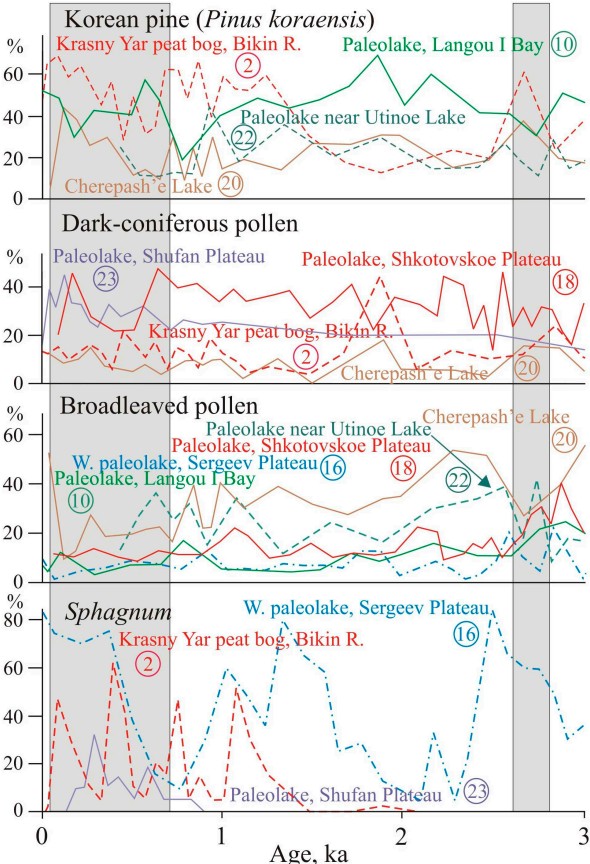

**Figure 4.** Comparison of results for pollen and *Sphagnum* remains from peat bog and paleolake sequences on the mainland. Grey bars indicate 2800–2600 year BP cold event and the LIA. Numbers of sites according to Figure 2.

In the valleys of Razdolnaya and Steklyanukha rivers, flood activity decrease led to the formation of buried soil, which is well expressed in sections of the high floodplain and the first fluvial terrace (Figure 5a). [14]C dates from 3320 ± 100 to 1520 ± 120 years ago were obtained from paleosol (Table 1). The diatoms typical for dry habitats (*Hantzschia amphioxys*, *Luticola mutica*, *Pinnularia borealis*, *P. obscura*) dominated in the paleosol. The presence of single hydrophilic diatoms is evidence of rare floods [65,66].

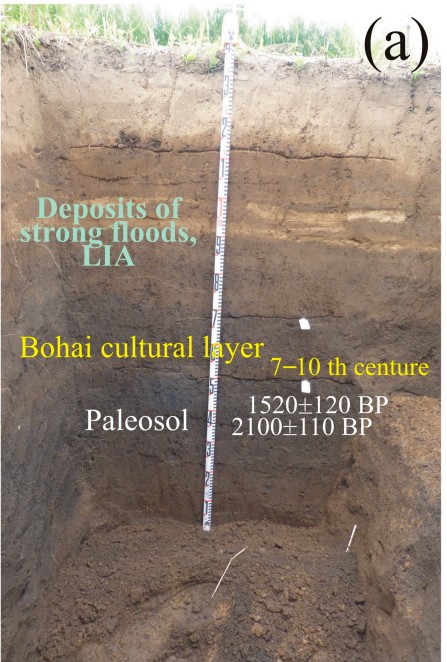 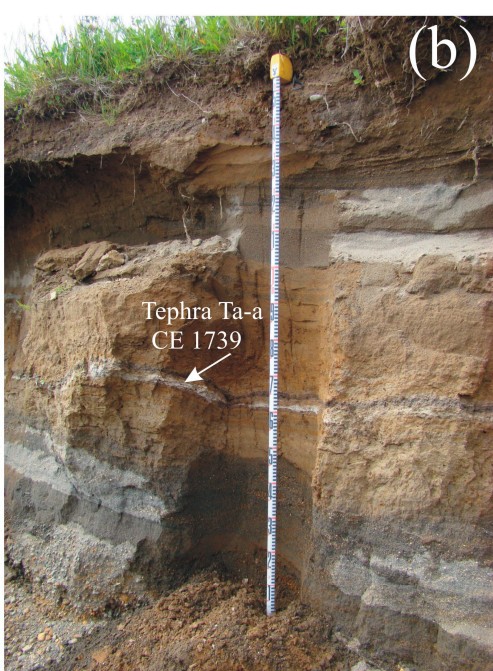

**Figure 5.** (**a**) Paleosol and deposits of strong floods on the floodplains of Razdolnaya (Suifun) River near Starorechenskoe Fortress, Southern Primorye (site 24 on Figure 2); (**b**) Deposits of the floodplain, Kurilka River, Iturup Island, Southern Kuril Islands—the deposits include marker tephra layers Ta-a (1739 CE) from the eruption of Tarumai Volcano, Hokkaido.

**Table 1.** Radiocarbon dating results and calibrated ages of the paleosol from flood-plain sequences and wetlands near the mountain lake, Primorye and Sakhalin Island.

| Location | Sample Number | Depth, m | [14]C-Age, Year BP | [14]C-cal Age, 2σ | Lab Number, LU- |
|---|---|---|---|---|---|
| Razdolnaya R. | 20/617 | 0.95–1.00 | 2110 ± 80 | 2100 ± 110 | 8854 |
| | 13/817 | 0.95–1.00 | 1610 ± 110 | 1520 ± 120 | 8856 |
| | 1/821 | 1.47–1.52 | 3060 ± 100 | 3240 ± 130 | 10,429 |
| | 2/821 | 1.72–1.78 | 3120 ± 80 | 3320 ± 100 | 10,430 |
| Steklyanukha R. | 1/720 | 0.35–0.40 | 2170 ± 110 | 2160 ± 140 | 9983 |
| Alekseevskoe Lake | 1/15,014 | 0.11–0.16 | 370 ± 70 | 410 ± 70 | 7709 |
| Yablochnaya R. | 1/322 | 0.29–0.31 | 2710 ± 80 | 2840 ± 90 | 10,755 |

On the coast of Southern Primorye [67], the cooling began 2870–2780 BP, as evidenced by decreasing concentrations of diatom valves and the appearance of arctoboreal species in the paleolake/peat bog section on the Amursky Bay coast near the Utinoe Lake (Figure 3). The quantity and variety of broadleaved pollen decreased sharply. Two dry episodes of 2780–2700 and 2610–2510 year BP led to a significant ecological reorganization of the diatom flora and a sharp increase in the content of soil species. The weak waterlogging separating these dry events occurred in cool conditions. The discovery of *Brazenia* seeds shows that the

reservoir warmed up sufficiently in summer. Planktonic *Aulacoseira laevissima* developed abundantly in the lake. This species was found among the dominant species during cooling 2110–1760 and 660–250 years ago in the sediments of Nizhnee Lake (565 m a.s.l.), Central Sikhote-Alin [49]. The swamp vegetation composition on the Amursky Bay coast indicates drier conditions 2780–2510 year BP, when a tree layer of shrub birch *Betula ovalifolia* developed on the swamp [67]. Development of ledum typical for pyrogenic successions is the evidence of frequent fires in drier conditions 2700–2510 year BP [64]. Sedge hummocks indicate a significant fluctuation in the swamp water level in dry seasons and during rains. Pollen spectra showed that the role of broadleaf trees in the forest vegetation on the coast of the Amursky Bay decreased during 2700–2610 year BP. A similar situation was recorded in the pollen spectra from the Utinoe Lake sediments [68]. Significant landscape changes associated with cooling were noted on the coast of the Boysman Bay, which is the more open part of the Amursky Bay, where alder-birch forests developed [69]. On the Murav'ev-Amursky Peninsula (the area near Cherepash'e Lake), a cooling was manifested in 2660–2340 year BP, broadleaf Korean pine forests spread (Figure 4), and Korean pine–spruce–broadleaved forests appeared in the near-top areas of the mountain ridges [70].

This cooling is associated with the migration of the representatives of the Krounovskaya Culture to the coast (2500–2200 year BP), which became more attractive than the inner parts of the river valleys suffering from severe droughts in early summer and floods in late summer–September [71].

On the coast of Eastern Primorye, an expansion of dark coniferous forests and the decreasing role of broadleaf trees were recorded in the sediments of Latvia Lake [72]. The temperature declined and signals of lower moisture are most clearly observed in the sediments of small reservoirs. On the coast of the Kit Bay, diatoms of the genera *Pinnularia* and *Eunotia* dominate in deposits of this age, many of them tolerate to low moisture and thus reflect the development of swamp processes [73]. A decrease in the paleolake watering due to a reduction in river runoff was established since 2600 year BP in the Langou I Bay [32].

### 4.1.2. Islands

In the south of Sakhalin, the cooling about 2750–2500 years ago was insignificant [30,51]. Diatom distribution in the section of the watershed peat bog of the Naiba River basin indicates that moisture decreased in 3220–1840 year BP, and extreme floods were rare events [50]. A decrease in moisture was also recorded in the development of a peat bog on the Terpeniya Bay coast where shrub birch became common in marsh vegetation [74]. In Western Sakhalin, a cooling is recorded in the peat bog section at the mouth of the Orokess River, where about 2610 year BP, oak–birch forests were replaced by birch–elm–oak forests on the coast, broadleaved forests decreased, and annual precipitation dropped (<600 mm). To the north, in the Sergeevka River valley in ~3000–2500 year BP, the role of spruce–fir forests lessened and birch forests, thickets of *Duschekia*, and larch forests became widespread. Under drier conditions, soil began to form on the peat bog [52]. In the Yablochnaya River valley, soil formation started on the peat bog ~2840 ± 90 year BP (Table 1) and there were no traces of strong floods. In the north of Sakhalin, a short-term cooling occurred ~2900 years ago [75].

In the Southern Kurils, cooling was also accompanied by moisture decrease, which was recorded in the development of lacustrine-swamp environments. On the coast of Zelenyi Island, paleolake depth declined in 2890–2500 year BP [76]. On Kunashir Island, the drier phase was shorter (~2590–2430 years ago), the content of Cyperaceae pollen sharply decreased, Asteraceae, Poaceae, Rosaceae, *Sanguisorba*, and *Thalictrum* became abundant, and a wet meadow began to replace the swamp near the paleolake. The diatoms are dominated by *Humidophila contenta*, *Luticola mutica*, *Hantzschia amphioxys*, *Pinnularia divergentissima*, which are typical for slightly moist areas (Figure 6). In cooler conditions, the participation of broadleaf tress in forests decreased [77]. The presence of *Scheuchzeria palustris* debris in some sections demonstrates cold environments [78]. The dry phase was also distinguished in the development of the coastal Glukhoe Lake (3300–2400 year BP),

which became very shallow and was replaced by a swamp. The rise in dark coniferous forests and the reduction in broadleaf species participation also reflect a regional shift toward cooler conditions [79]. Proportions of arctoboreal diatoms increased in the paleolake sediments of Kunashir and Shikotan islands [77,80] that well correlated with the cold latest Jomon stage of Japan [81].

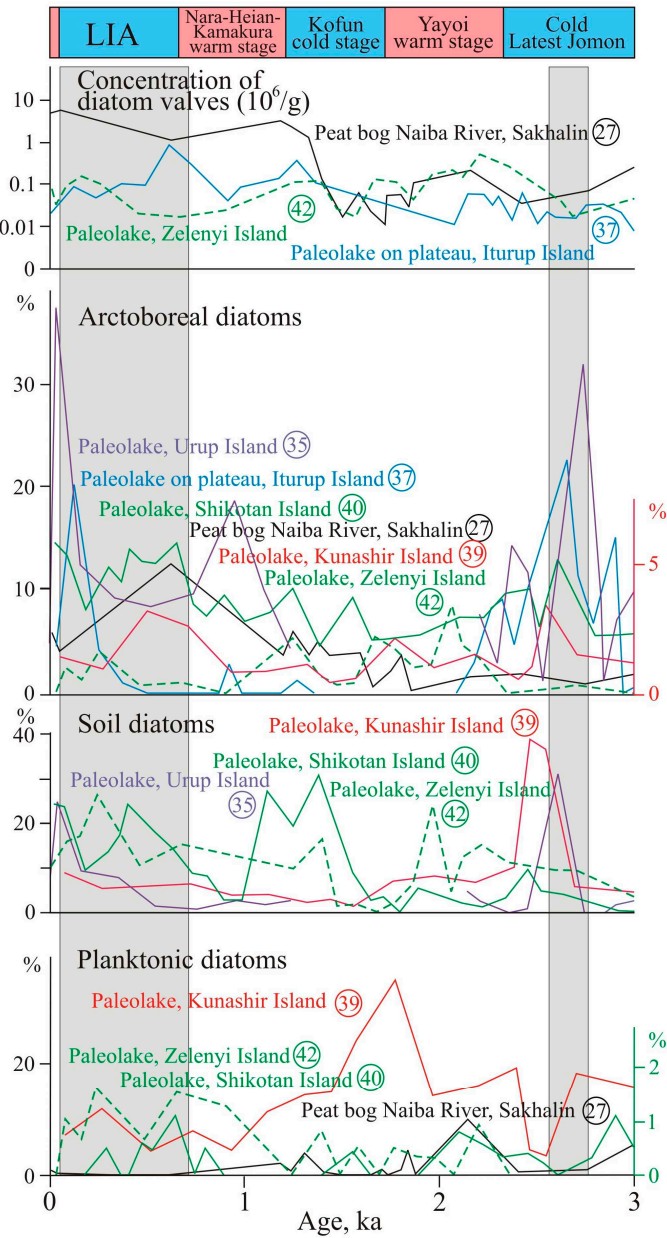

**Figure 6.** Comparison of results for diatom proxies from peat bog and paleolake sequences on the islands. Japan cold and warm stages in cal. year BP [81]. Grey bars indicate 2800–2600 year BP cold event and the LIA. Numbers of sites according to Figure 2.

In the paleolake sediments on a plateau of Iturup Island (400 m a.s.l.), a sharp increase in arctoboreal diatoms 2920–2570 years ago indicates cold conditions (Figure 6). There was an alternation of long dry phases and more humid periods ~2800–2650 year BP; sometimes, the lake dried up almost completely. In the conditions of desiccation, a large amount of heather bushes developed on the swamp. More open areas appeared in the mountains. The proportion of dwarf pine pollen (*Pinus pumila*), an indicator of stable and heavy snow cover [82,83], is insignificant (Figure 7), which indirectly indicates that there were no heavy

snowfalls. An increase in allochthonous broadleaved and *Cryptomeria* pollen transferred from the southern islands was noted, and the active supply of oak and elm pollen from low relief levels (Figure 7) indicates cyclogenesis activation in spring and summer [84]. On the coast, a drop in the level of Lebedinoe Lake was noted for 2800–2650 year BP, the areas of broadleaf forests decreased, and alder thickets began to develop around the lake [85]. On Urup Island, cooling with a decrease in moisture at ~2840–2410 year BP was recorded in the paleolake sediments in the Osma Bay coast [86]. A reduction in diatom species diversity was noted, the proportion of plankton diatoms sharply decreased (≤1.3%), benthic species began to predominate (up to 88%), rheophilic species disappeared, the content of soil diatoms (Figure 6)—mainly *Pinnularia borealis*—rose significantly (up to 29.6%), the aerophilous *Diatomella balfouriana* appeared, and the participation of acidophiles and halophobes increased. These facts testify to the degradation of the lake and active swamping.

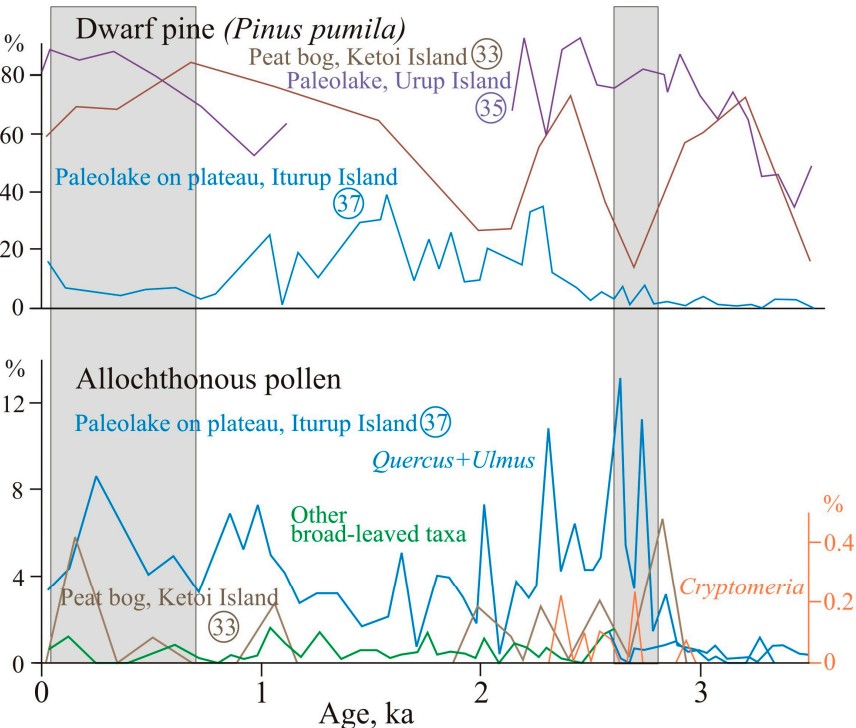

**Figure 7.** Comparison of results for pollen from peat bog and paleolake sequences on Kuril Islands: *Pinus pumila* pollen is an indicator of snowfall and winter cyclogenesis activity; allochthonous pollen from sediments of paleolake on mountain plateau (400 m a.s.l.) on the Iturup island, including *Quercus* and *Ulmus* pollen, were windblown from lower landforms; other broadleaved pollen (*Carpinus*, *Tilia*, *Juglans*, *Corylus*, *Fagus*) and *Cryptomeria*, that were transported from the south of both the Kurils and Japanese islands, are indicators of cyclone activity during spring–summer. Grey bars indicate 2800–2600 year BP cold event and the LIA. Numbers of sites according to Figure 2.

In the Central and North Kurils, cooling led to the expansion of tundra areas. On Matua Island cereal-forb meadows developed and dwarf alder occupied limited areas. On Rasshua Island, slightly moistened swamp began to replace the paleolake in the ancient caldera; the typical soil species *Pinnularia borealis* became dominant among diatoms (up to 60%). Cooler conditions resulted in widespread communities of *Pinus pumila* with *Selaginella selaginoides*, the species combination indicating heavy snowfalls [87]. On the islands where volcanoes erupted actively in the late Holocene, dwarf pine community areas decreased (i.e., Ketoi Island, Figure 6) or *Duschekia* prevailed in the shrub communities. Distinguishing the climatic signal here is problematic due to active volcanism and the formation of thick

tephra sheets, strongly affecting the hydrological regime. Allochthonous broadleaved and dark coniferous pollen, windblown from the southern islands, were found.

*4.2. Manifestation of the LIA*

4.2.1. Mainland

The LIA was characterized by the most dramatic and profound climatic changes during the Holocene [28,88–90]. Like other regions of Asia [3,11,81,91–93], the Southern Russian Far East had an unstable climate, with changes not only in the temperature, but the humidity as well.

The transition from the Medieval Warm Period (MWP) to the LIA was marked by frequent hydroclimatic changes. Cooling in the Central Sikhote-Alin began to manifest in the middle of the 12th century. Increase in arctoboreal diatoms in the microflora of the Izyubrinye Solontsi Lake occurred in 1150 CE. There was a short period of moisture decrease (1110–1150 CE), when the Nizhnee Lake was completely overgrown (Figure 8). Findings of micro-charcoals show that fires occurred during long dry seasons [49]. On the Shufan Plateau, the flooding was overlain by a short-term (1030–1110 CE) signal of moisture decrease, identified by the presence of soil diatoms, which coincides with one of the cold anomalies in China [94]. This event is consistent with the Oort Minimum (1010–1080 CE). It became colder at ~1160 CE. A cold relapse ~1150 CE is also known in Europe [91] and China [94]. In the foothills of the Sikhote-Alin, this dry phase was not manifested. In the lower reach of the Bikin River valley, the mires were flooded and severe floods took place during 1000–1190 CE. The data obtained for the south of Primorye are well compared with those for the nearby areas. In the north of the China Plain, a period of strong droughts occurred during 1120–1210 CE [93]. According to the chronical, 1180–1320 CE was a period of frequent droughts in the south of the Korean Peninsula [9].

At the beginning of the LIA, mountain lakes and swamps were watered. The role of *Sphagnum* mosses increased on mountain and valley mires (Figures 4 and 8) [35,49,60,63]. The water content of rivers and the frequency of floods increased sharply. In the river valleys, sandy loam and loam accumulated with no traces of long breaks on the flood-plains (Figure 5a). Cold and humid conditions were overlain by short-term phases of moisture decrease. Near the Solontsovskie Lakes, conditions became cold and humid in ~1190–1290 CE. The lake basins were occupied by *Sphagnum* swamps. Nizhnee Lake was the most sensitive to hydroclimatic changes [49]. The highest watering was 1290–1700 CE, with a short-term decrease in the lake level ~1570–1600 CE. The next significant flooding of the lake was in the first half of the 19th century. The arctoboreal diatom peaks show colder episodes in 1260–1290 CE, 1360–1390 CE, and 1480–1510 CE. All three cold episodes are recorded by negative temperature anomalies in China [94,95]. Apparently, the first cold event corresponds to sharp short-term declines in solar activity before the onset of the Wolf minimum [96], recorded and dated in tree rings in Japan from 1261 to 1280 CE [97,98]. The second event is close to the Wolf minimum (1280–1350 CE). At this time, there are signs of a decrease in moisture in the development of lakes: the overgrowing of lakes was more active, and cladocera disappeared. The third cold event corresponds to the second half of the Spörer Minimum [96,99]. Close temporal estimates of the manifestation of the Spörer Minimum of 1455–1510 CE were recorded in Japan based on the $^{14}$C content in the annual *Cryptomeria japonica* rings, which were 712 years old [97,98]. The decrease in arctoboreal diatoms is evidence of relatively warm episodes in 1290–1330 CE, 1420–1450 CE, and 1510–1540 CE. Compared to China [94], the boundaries of these episodes are relatively shifted, which can be explained by inaccuracies in the age model. These events are corre-lated with temperature increases in the Northern Hemisphere [100,101]. Europe had one of the warmest winters in 1290 CE, and the warm summer period was around 1550 CE [91]. Warming around 1450 CE manifested itself in the region of the Middle Amur [100]. In China, a warm episode is recorded around 1520–1560 CE [92]. An increase in planktonic diatoms indicates the flooding of the Nizhnee Lake 1290–1570 CE [49].

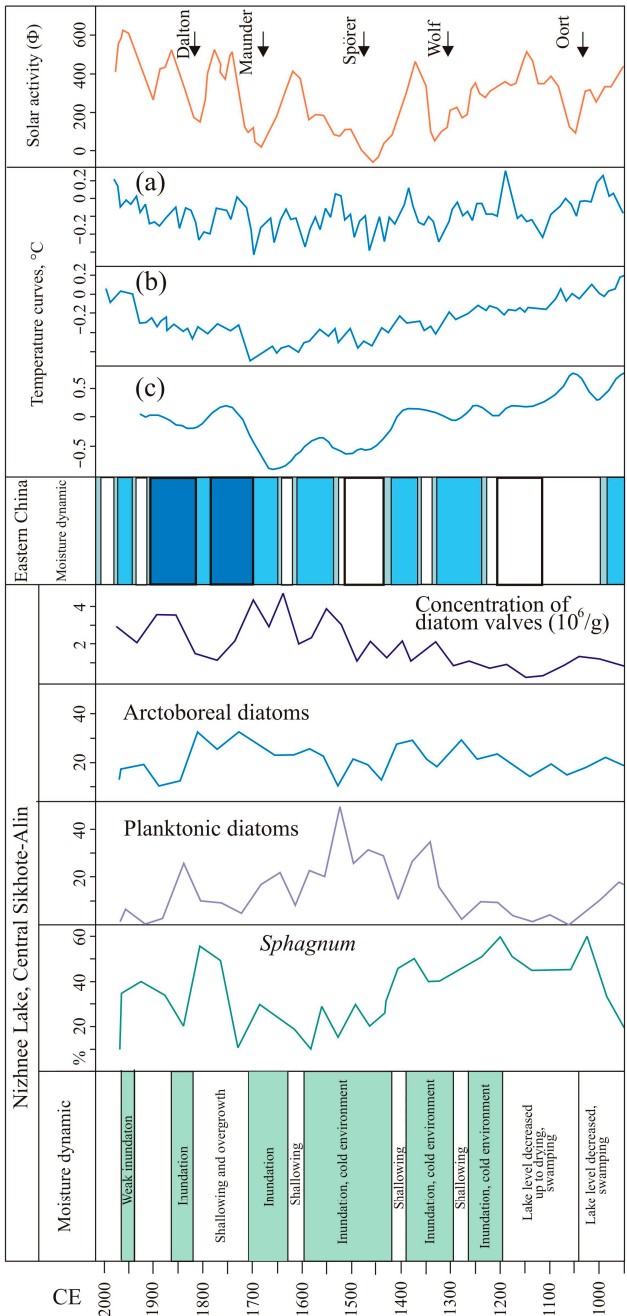

**Figure 8.** Evolution of Nizhnee Lake (site 9 on Figure 2) and changes of diatom proxies during the last millennium. Solar activity fluctuations reconstructed based on [10]Be measurements in Antarctica ice [96]. Temperature curves: (**a**,**b**) reconstruction of temperature anomalies for the Northern Hemisphere from average during 1951–1980 CE [89] and 1961–1990 CE [90]; (**c**) Chinese temperature anomalies (sigma units) [94,95].

In the second half of the LIA, a cooling of 1570–1600 CE is distinguished by an increase in arctoboreal diatoms, which is well comparable with the cooling peak in China [94]. Nizhnee Lake became shallow, a grass layer developed in the surrounding swamp, and the participation of sphagnum mosses decreased. An increase in arctoboreal diatoms ~1710 CE records cold conditions at the end of the Maunder Minimum (1645–1715 CE) [88,96]. It became drier and a variety of green mosses began to develop on the drained swamp shore. About 1600–1660 CE there was a fire and the mineralization of peat increased slightly. Precipitation decrease in Southern Sikhote-Alin 1645–1680 CE was also recorded in tree



rings [39]. A cold episode stood out on paleotemperature curves for the Northern Hemisphere [88,89,99]. Cold conditions existed in China in the 16–17th centuries [93,102,103]. The cooling was also significant in the Middle Amur basin (the area of Blagoveshchensk City) where the average seasonal and annual temperatures were 2 °C lower than the present ones, which is the minimum for the last 2000 years, and the amount of annual precipitation decreased by 100 mm [100].

On the western macroslope of the Sikhote-Alin in the Bikin River basin, a sharp increase in arctoboreal diatoms (up to 30%) in the peat bog sections was also established in 1270 CE, which coincides with a short cooling period in China [94]. Under the conditions of reduced water flow, no strong floods and fires occurred in a frequent manner (1230–1270 CE) [34]. A dry phase is also recorded in the continental part of the Amur basin [31]. The Bolshaya Ussurka River basin became drier 1290–1360 CE [35], corresponding to the cold episode in China [94]. At the same time (1305–1400 CE), in the Bikin River basin, the intensification of floods was noted. An outbreak of *Sphagnum* development was recorded for 1380–1440 CE. Green moss *Mylia anomala*, an arctoboreal circumpolar species often found in blanket bogs on the Tatar Strait coast along with other tundra plants, appeared in the mire vegetation of the Lower Bikin River [34]. The watering of swamps and an increased frequency of floods were observed for 1460–1530 CE. In the Bolshaya Ussurka River basin, the opposite situation was noted: a decrease in moisture during 1420–1490 CE occurred, but with frequent floods [35]. Climatic conditions in this part of the Sikhote-Alin western macroslope were closer to those of Eastern China, with 1430–1520 CE negative temperature anomalies [94], and frequent prolonged droughts were noted [93]. A dry period of 1420–1520 CE also stood out in South Korea [9]. There was a pronounced cooling in China [94]. The fall in flood activity of the Bikin River was recorded during the cold phase 1530–1730 CE. At the same time, during 1550–1610 CE, frequent floods occurred in Bolshaya Ussurka River basin [35]. Since 1540 CE, climate became wet in East China [93]. According to the Annals of the Joseon Dynasty (1392–1863 CE), flood frequency also increased in Korea [9]. An outbreak of Sphagnum moss (mainly *Sphagnum magellanicum*) on the mires of the Middle Bikin River was recorded in 1610–1660 CE [63]. The role of broadleaf species in the forests on the surrounding mountain slopes was reduced.

In the main watershed of the Southern Sikhote-Alin, the flooding of swamps under colder conditions occurred in 1240–1380 CE, and fields of open water appeared. Under conditions of high moisture in the near-top flattened part of Olkhovaya Mountain, Alekseevskoe Lake was formed in the permafrost-nival depression (Table 1). It is the only lake in the mountain tops. The lake formation was most likely facilitated by an increase in annual precipitation in the 15–16th centuries, with high winter precipitation. For East China, from the end of the 15th to the beginning of the 16th centuries, intense snowfalls and the formation of a thick snow cover were typical even in the modern zone of the subtropics [89]. In the 17th century, the watering of Muta peat bog decreased. A slight warming during the 18th century was seemingly accompanied by an increase in precipitation. At the time, the Alekseevskoe Lake became deeper. The lake had atmospheric supply and one of the likely factors for the level rise was an increase in the number of snowfalls. Warming was recorded in Southern Sikhote-Alin according to tree-rings data [38] and in the southernmost coast of Primorye according to pollen records [104]. Korean pine tree-ring-base records in the Chaibai Mountains (Northeast China) indicate a relatively warm climate in 1750–1783 CE [105]. In East China, one of the wettest periods with severe floods was 1700–1790 CE [93]. Relative warming (1701–1780 CE) was noted in the chronicles for the North China Plain, followed by cooling, which continued until the end of the 19th century [103]. This cooling coincided with the Dalton minimum (1790–1820 CE) [88,96].

On the Southern Sikhote-Alin plateaus in lacustrine sediments, the maximum of arctoboreal diatoms was recorded for 1365–1480 CE (Shkotovskoe Plateau) and before 1560 CE (Sergeev Plateau) (Figure 3). An increase in bog watering led to the suppression of the tree layer and the development of sphagnum bogs with predominant *Sphagnum fuscum*, *S. magellanicum* and abundant cranberry. Among the green mosses, the arctoboreal

*Aulacomnium turgidum* appeared [60]. This coincides well with negative temperature anomalies [94]. During the LIA, sphagnum-herbal swamp replaced the paleolake on the Shufan Plateau [33]. The participation of sphagnum mosses increased in colder episodes (1370–1440 CE and 1650–1710 CE) (Figure 4), which were also well expressed in China [94]. In the moss cover, circumpolar *Sphagnum cuspidatum* and *S. contortum* began to occur along with *S. fallax*, and cotton grasses (*Eriophorum russeolum*, *E. scheuchzeri*) typical for northern taiga, forest tundra, and tundra also appeared. The inundation of the lake basin decreased in 1590–1750 CE, as evidenced by decreasing planktonic diatoms and increasing epiphytes. The peak of shrub birch pollen (*Betula ovalifolia*) indicates cold and less humid conditions. The areas presently occupied by the species in the south of the region are a LIA relict [33]. This dry episode correlates well with cold temperature anomalies in China [94]. Data on changes in the Khanka Lake level reflect moisture change at the end of LIA. The level fluctuated greatly: high in the middle of the 18–19th centuries [106,107] and beginning to fall in 1866–1893 CE, according to first settlers' observation [108].

4.2.2. Islands

In Southern Sakhalin, the cooling of the 12–14th centuries was well expressed as the climate became drier compared to MWP. Larch forests expanded, and frigid shrubs with a predominance of shrub birch became common on swamps [30]. In the 14–15th centuries, a warming was noted; the area of dark-coniferous forests with a participation of broadleaf species increased. The period between the 16th and the first half of the 19th centuries was the coldest, with a decrease in moisture, and landscapes similar to the forest-tundra of the Northern Sakhalin Island developed in the south [51]. In Eastern Sakhalin, the 400-year-old tree-ring records show that the coldest years were 1680–1710 CE, which coincides with the Maunder Minimum [37]. In the northwest of the island, short-term cold and dry events were identified at about 1050–900 CE and 1450 CE [75]. Diatoms of the watershed peat bog in the Naiba River basin showed an increase in arctoboreal diatoms (Figure 6), including aerophile *Pinnularia lagerstedtii*, which often inhabits wet mosses, and *Navigeia paludosa* appeared, a species which lives on moist soils and in biotopes with a tendency to temporary dehydration. There were no strong floods [50].

On the Kuril Islands, the cooling was well expressed. Quantitative estimates of temperature changes over the past 400 years based on geothermal and tree-ring data for Kunashir Island showed that the temperature of the Earth's surface in the 17–19th centuries was significantly lower (up to 2 °C) than at present [36]. The reconstructed course of temperature changes is in good agreement with the data for the Japanese Islands. Well-defined temperature minima were around 1700, 1745, and 1780 CE, which coincide with the tree-ring data on Kunashir Island [109]. Northern Japan was characterized by heavy snowfalls [81]. The cooling in the Kuril Islands could have been enhanced by the activation of the cold Oyashio Current and the shift of warm Soya Current 800–150 years ago [110].

In the Southern Kurils, the cooling led to a reduction in broadleaf forests and a decrease in broadleaf species in conifer–broadleaf forests [111]. On Shikotan Island, this cooling was most likely the cause for the disappearance of oak, resulting in an expansion of the areas occupied by dark coniferous and birch forests. The constant presence of allochthonous pollen in the pollen spectra indicates active cyclogenesis in the Southern Kurils region (Figure 7).

An increase in moisture ~AD 1290–1410 was recorded in the paleolake sediments on a mountain plateau in Central Iturup. The concentration of diatom valves increased (Figure 6), an outbreak of *Eunotia exigua*, which is typical for swamp waters, was noted, and the proportion of *Pinnularia borealis* was significantly reduced [84]. In the mountains (Figure 7) and at low relief levels, the role of dwarf pine grew [84,85,112]. On the coast, the areas occupied by relic oak forests decreased and sedge communities became widespread [111]. Humidification has decreased over the past 180 years, and the number of diatoms characteristic of poorly moistened moss bogs is increasing [85].

The study of the paleolake sediments in the south of Urup Island showed that broadleaf species disappeared from forest vegetation ~1350 CE. A decrease in temperatures at the time is noted on the global temperature curve [113]. In the south of Urup Island, forest vegetation degraded and heaths became widespread [86]. In the north of the island, cooling led to the expansion of dwarf pine [114].

In the Central Kurils, cooling manifested clearly since the 13th century and led to the expansion of tundra landscapes [115]. The proportion of sphagnum and green mosses increased in swamp vegetation. An increase in moisture and thickness of the snow cover is evidenced by a large number of *Selaginella selaginoides* spores in the pollen spectra, which was recorded in the peat bog section on Matua Island. The increase in arctoboreal diatoms *Pinnularia divergens*, *Brachysira serians*, and *Frustulia rhomboides* testifies to cold conditions.

On all the islands, the cooling was accompanied by a low-amplitude regression, which contributed to dune formation (Figure 9). The lowering of the sea level is evidenced by the numerous finds of peat bogs on the bench found on the islands of the Lesser Kuril Ridge, with interlayers of volcanic ash Ko-c$_2$ (1694 CE) and Ta-a (1739 CE) at the base, which indicates that the regression began at the end of the 16th century (Figure 10). The sea level decreased by at least 1 m. The similar estimates were obtained for the Japanese Islands (the Edo regression) [81]. One of the factors causing the active development of eolian processes was the strengthening of the wind regime, especially in winter [114]. The LIA was characterized by strong floods, as evidenced by thick sections of floodplain sandy loam in the lower parts of the rivers. On Iturup Island, volcanic ash Ta-a (1739) was found in sections of the LIA floodplain sandy loams (Figure 5b). Under conditions of frequent rains, slope processes were activated; sections of slope sediments of this age are exposed in erosion escarps in the coastal zone.

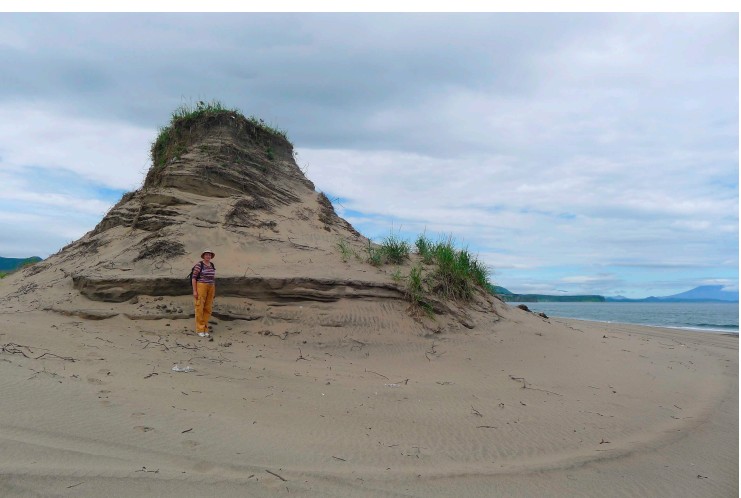

**Figure 9.** Dune of the LIA on the Pacific side of Kunashir Island, Southern Kuril Islands.

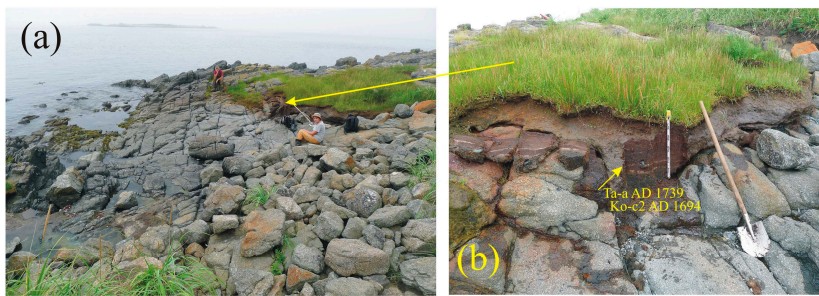

**Figure 10.** (**a**) Peat bog of the LIA (site 43 on Figure 2) on an uplifted bench of Tanfiliev Island, Lesser Kuril Ridge; (**b**) peat bog section with marker tephra layers Ta-a (1739 CE) and Ko-c2 (1694 CE) from the eruption of Tarumai and Komagatake volcanoes, located on Hokkaido Island.

## 5. Atmospheric Circulation Patterns

The analysis of the available data showed that two late Holocene global cooling events of a similar temperature amplitude, which were manifested clearly in the Southern Russian Far East, differed in moisture. As in the analysis of hydroclimatic changes in the last millennium in China [12], in the south of the Russian Far East, different reactions of moisture/precipitation to similar patterns of temperature changes during two cold events are clearly manifested. Studies of recent climate changes [15,43,116] and paleo-events [9,11,12,25,26,95,117–119] show that humidification at the continental margin and islands strongly depends on the processes occurring in the ocean–atmosphere system. In order to understand the mechanism controlling moisture variations during two cold events of 2800–2600 year BP and the LIA, as in other regions of East Asia, we propose to consider the change of the atmospheric circulation, the intensity and position of the atmosphere centers of action, the intensity of the summer and winter monsoons, and characteristics such as surface sea temperature (SST) in the western part of the tropical and subtropical Pacific, intensity and frequency of ENSO, i.e., processes that control the intensity of cyclogenesis, and the trajectories of deep cyclones and typhoons that cause precipitation [43,47,120].

### 5.1. Cold Event 2800–2600 Year BP

Global cooling 2800–2600 year BP coincides with one of the well-defined minima of solar activity, recorded by $^{10}$Be in ice cores of Greenland (Figure 11a) and $^{14}$C in the annual rings of trees of known age [3,6,8]. This grand solar minimum coincided with the Hallstattzeit cold epoch (2700–1350 year BP), a time that was wetter in Western Europe and drier in Eastern Europe [121]. The average annual temperature in the south of the Far East was 1.5–2 °C lower than present, and in the southernmost of Primorye, 0.5–1 °C lower than present; on Sikhote-Alin mountain tops and on the Shantar Islands, the cooling was more pronounced at 2 °C lower than present. Winter temperatures were lower than present: Primorye, 1–1.5 °C; Sakhalin, 1.5 °C; and the Kurils, 1 °C. Summer temperatures also decreased, with the following temperatures lower than present: Primorye, 1.5 °C; Eastern Primorye up to 2 °C; Sakhalin, 1.5 °C; and the Kurils, 1 °C [122]. Everywhere in the south of the Far East, the cooling was accompanied by a decrease in moisture (Figures 3 and 6). Annual precipitation decreased by up to 50 mm lower than present on Primorye and the Kurils, and up to 100 mm in the northernmost Primorye, Sakhalin, and the Shantar Islands [122]. In general, these changes occurred synchronously on the mainland and the islands, with a slight difference in the duration of the dry period in the southernmost mainland [76], which lasted the longest in the mountains and river valleys from ~3320–3050 year BP until MWP [33]. In river mouths and on the sea coast, drier environments settled since 2950 year BP, and the driest period was 2780–2510 year BP. A weak watering ~2700–2610 year BP was recorded only in one section on the coast of the Amursky Bay [67]. In the Sikhote-Alin and in the foothills, moisture decreased in 3080–2900 year BP, and the driest conditions occurred in 2760–2735 year BP [63]. In Southern Sakhalin, moisture decreased ~3220 year BP, and the drier period was during 2840–2500 year BP (Figure 6). In the Southern Kurils, on the coast and in the mountains, it became colder and drier ~2920–2890 year BP, and the driest period was 2590–2430 year BP [77]. It also became colder and drier in the Central Kurils (Figure 6). On the Shantar Islands, a cooling and a decrease in moisture were recorded for 2830–2695 year BP; in this part of the Okhotsk Sea, high storm activity was observed ~2960–2570 year BP, and winter cyclogenesis became more active [123]. The cold event corresponds with Bond event 2 (Figure 11).

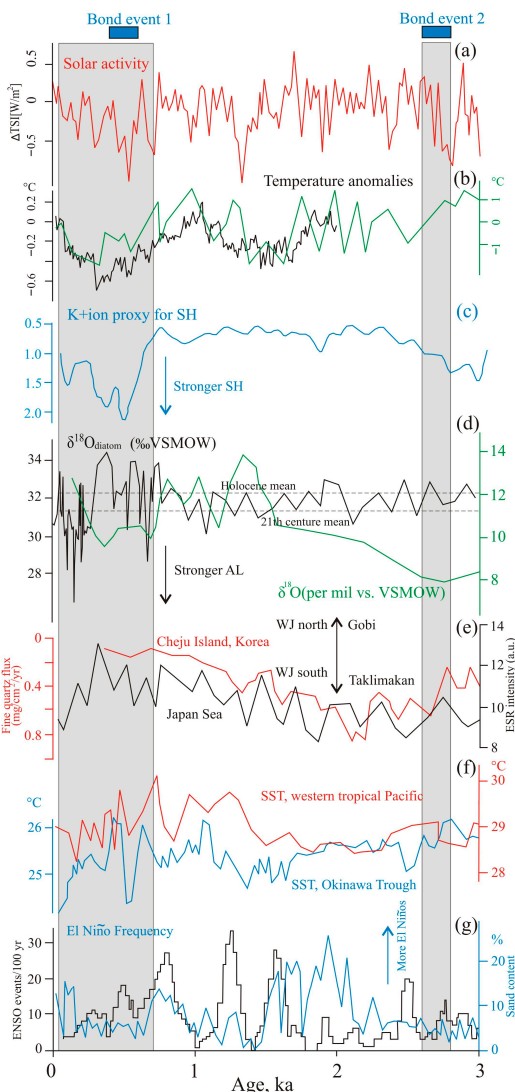

**Figure 11.** Compilation of selected proxy records of sun activity, palaeotemperatures, SH and AL intensity, WJ shift, SST of western subtropical and tropical Pacific, and El Niño frequency for the past 3000 years. (**a**) Solar activity fluctuations reconstructed based on [10]Be measurements in polar ice [6]. (**b**) Black line indicates relative decadal mean temperature variations (°C) for 1961–1990 CE with estimations of extra-tropical Northern Hemisphere (90–30° N) [90]. Green line—the reconstruction of annual temperatures for the Amur River region from modern average temperature [100]. (**c**) Gaussian-smoothed (200 year) GISP2 potassium (K+; ppb) ion proxy for the Siberian High [4,20]. (**d**) Time series of δ [18]O value as proxy for the AL intensity of δ [18]Odiatom value from Heart Lake on Adak Island and Aleutian islands [24]; δ[18]O from bulk peat on the Kenai lowlands in South–central Alaska [23]. (**e**) Dust proxy for WJ shift: black line—ESR signal intensity of quartz in sediments from the Japan Sea [13,22]; red line—fine quartz flux deposited at Cheju Island, Korea [124]. (**f**) SST of western subtropical and tropical Pacific: blue line—alkenone-based SST from site ODP 1202B, Okinawa Trough [125]; red line—Mg/Ca-palaeoSST records with results from site MD81, tropical Pacific [126]. (**g**) El Niño frequency: black line—time series of ENSO events, Laguna Pallacoacha, Ecuador [17]; blue line—content of sand in the sediments of Junko Lake, Galapagos [27]. Blue bars indicate Bond events [1,2]. Grey bars indicate 2800–2600 year BP cold event and LIA.

The cooling was manifested in East Asia and marginal seas and was synchronous with a cold and dry event ~2800 year BP in China (Zhou–Han Dynasty cold period) [125]. Paleoclimatic proxy records in the Southern Russian Far East (Figures 3–6) are in good agreement with the late Holocene summer monsoon decline in northeastern China. A decrease in

precipitation was recorded in the pollen proxy from the sediments of Jingbo Lake [127,128], the carbon-isotope and geochemical records from the Hani Peat section [14,21,129,130], and the grain-size proxy from the peat section in the Changbai Mountains [131]. The $\delta^{18}$O records of stalagmite from Chinese caves indicate a weaking monsoon event between 2900 and 2400 year BP [132]. Based on stalagmite $\delta^{18}$O record, a weak EASM event 2700 years ago was identified on the Korean Peninsula [133]. It was also dry on the coast of Korea ~2800–2300 year BP [11]. The latest Jomon stage (2820–2350 year BP) was prominent in the Japanese Islands, its beginning being the coldest since the Holocene Optimum and was characterized by an increase in winter precipitation [81,134].

The available paleoenvironmental material for the northern and western parts of the Pacific Ocean makes it possible to reconstruct in general terms the features of the atmospheric circulation for this cooling. The $\delta^{18}$O$_{diatom}$ records of moistening in lacustrine sediments from Adak Island and $\delta^{18}$O records of cellulose from the peat bog of South Alaska (Figure 11d) show that the AL was strong at this time and shifted eastward [23,24]. This is also confirmed by data on the northeastern Pacific margins [25]. An increase in the AL intensity is associated with the data on ice coverage increase in the Okhotsk Sea and the Bering Sea; a slight increase in sea-ice-related diatoms during this period was recorded in the sediments of the southwestern part of the Okhotsk Sea [135]. In other areas of the Okhotsk Sea, a cold event is distinguished ~3200–2600 year BP [29]. In the late Holocene, NPH shifted southward and/or weakened [25].

The position and intensity of the SH is estimated by a change in the trajectory of WJ (Figure 11e). The trajectory was reconstructed using the ratio of quartz transported by dust storms from the Mongolian Gobi and Taklimakan deserts to the Japan Sea and was recorded by electron spin resonance signal intensity. A sharp decrease in the proportion of quartz from the Gobi Desert in dust ~3500–1500 year BP is interpreted as a decrease in the frequency of strong dust storms arising due to cold dry air transport from Siberia [13]. The flow of fine quartz brought by dust storms to the maar lake sediments on Jeju Island from the Taklimakan desert also increased, which can indicate a longer stay of the WJ of the westerlies on the south side of the Tibetian Plateau in 3100–2500 year BP [124]. During the period of 3500–1500 year BP, humidity/precipitation tended to increase at the northwestern EASM margin, while the Northeast China, North China, and the Yangtze River Basin were drier [13].

Comparing sea ice records of the Okhotsk and Bering Seas with dust records [13] shows that periods of high sea ice extent in both seas may correspond to frequent and/or strengthened winter–spring storms in the Mongolian Gobi Desert, allowing for the suggestion of eastward migration of the center of SH [135]. Using this conclusion, we can assume that during the period of weakening winter–spring storms in the Mongolian Gobi Desert 3500–1500 BP [13], the SH center shifted westward. The major ion series (analysis of the continental-sourced K) in an ice core from Central Greenland shows that the influence of SH increased at the time (Figure 11c) [4,20]. Despite the fact that the anticyclone was stronger, its effect weakened in the Far East during the cooling of 2800–2600 year BP. The weakening of the SH at this time is also confirmed by the synthesis of paleoecological data from the lake records of Kamchatka [136].

The observations of the second half of the 20–21st centuries [44] show that when SH shifted to the west, the frequency of cyclones have increased everywhere, but their intensity decreased. The SH mode location is related to the character of transport in the troposphere; it occupies the western position during meridional process development [137]. At the same time, cyclogenesis develops due to shallow western cyclones that have a predominantly meridional orientation, often reaching the Okhotsk Sea [44]. Most of the winter cyclones move to the Aleutian Islands [46].

Such interrelation between the intensity of the atmosphere centers of action and the cyclogenesis development explains the low amount of winter precipitation in the Southern Far East during the cooling of 2800–2600 year BP. The trajectory of deep southern cyclones, which caused heavy snowfalls, must have shifted to the east. Based on pollen proxy,

intense snowfalls occurred at the time in the Central Kurils [111], where the maximum intensity of southern winter cyclones is currently observed [46]. More abundant snowfalls compared to the previous Holocene period were also recorded on the Shantar Islands [123]. It is possible that the frequency of winter cyclones which originated over the Japan Sea increased, because the temperature contrast between land and ocean increased, which favors winter cyclogenesis. The western cyclones from Transbaikalia usually intensify here as well, filled up in the middle reaches of the Amur River and transforming into a new cyclone over the Japan Sea [46]. There is evidence that the Tsushima Current in the Japan Sea was active at the time [138], with the current possibly increasing the temperature contrast between land and sea. Cooling leveled the effect of intensification of the Soya Current (branch of the Tsushima Current) to the terrestrial environment in the southern part of the Okhotsk Sea [139]. Possibly due to the influence of the warm current, the cooling in the south of Sakhalin Island was insignificant [30].

At present, when the SH is located in the western position, the prolonged development of meridional processes in winter, as a rule, increases the duration of the winter type of circulation in spring [48]. When the SH is weakly expressed, the FEL is weak and shifts to the southwest, where numerous shallow western cyclones emerge from this center; the situation continues in summer, and OH is also weakened in this case [44]. In such a situation, the outflow of cold moist air to the northern part of the Japan Sea was most likely less significant. The blocking role of the anticyclone also decreased, which could also affect the trajectories of summer cyclones bringing precipitation. Evidence of cyclogenesis activation is in the increase in allochthonous broadleaved taxa and *Cryptomeria* pollen transferred by bioaerosol from the southern islands (Figure 7).

In general, the shift of the mode of the atmosphere centers in opposite directions and the pressure gradient between the continent and the ocean, as in modern conditions [45], could result in weakening both winter and summer monsoons. Such changes in atmospheric circulation could be one of the reasons for the decrease in precipitation. At present, cycles (~20 year periodicity) are distinguished when both centers occupy extreme western and eastern positions, the latest of such a period being the beginning of the 2010s [140].

Atmospheric circulation anomalies and the development of summer cyclogenesis are strongly affected by SST in the western Pacific tropical zone (Figure 11f) where the main centers of typhoons are located [47,118,141,142]. During the cooling of 2800–2600 year BP, SST in the western Pacific tropical zone and in the Okinawa Trough was ~0.5–0.7 °C lower than the present [125,126,143]. The SST long-term variability in the Pacific tropical zone was strongly affected by ENSO, which caused anomalies in tropical and extratropical cyclogenesis and the monsoon system of East Asia [11,15,25].

Data on the tropical proxy show that the peak of ENSO activity began precisely since 2800 year BP (Figure 11g) and it became the main climatic factor for the northeastern [25] and northwestern parts of the Pacific region [9,11,118]. These processes influence the intensity of the Kuroshio Current and its branches, as well as the dynamics of the summer monsoon [94]. The decrease in SST is associated with the activation of El Niño [17,125,143], which led to a weakening of the summer monsoon, [11] and, apparently, a significant weakening of cyclogenesis. As in modern conditions [144], during strong El Niño years, in the western Pacific Ocean, warm water masses could shift to the east, and the center of tropical cyclogenesis most likely shifted as well, and typhoon tracks could then shift to the Japanese islands and move into the ocean. An increasing frequency of extreme floods during 2900–2500 year BP is recorded in the sediment of Lake Suigetsu in Central Japan, where the main typhoon tracks passed [145].

Similar results were obtained when modeling atmospheric precipitation under conditions of a 1 °C decrease in SST [146], Primorye and the South Kuril Islands should fall into the area of decreasing moisture, along with Sakhalin and the Okhotsk Sea. During the El Niño years, the AL and OH strengthened and the NPH and the Asian Low weakened, which led to an increase in extratropical cyclogenesis [47]. Over the northwestern Pacific and the Far East, there is a weakening of zonal flows and an increase in the fre-

quency of intrusions of cold air masses from the north, which causes a decrease in air temperature [15,147].

At present, prolonged low water levels and abnormal droughts in the continental part of the Southern Russian Far East occur during the years of strong long-term classical El Niño [15]. The decrease in precipitation in the cold and warm seasons is associated with the intensification of the winter monsoon and the weakening of the summer monsoon; typhoon trajectories in such years shift to the east, away from the continent, and as a rule do not reach Primorye [15]. In the cold event of 2800–2600 year BP, the Subarctic Front was ostensibly located south of its modern position, and the number of cyclones passing over the South Kuril Islands decreased. The eastward shift of the trajectories is indirectly evidenced by traces of strong paleotyphoons in the south of the Japanese Islands (Kamikoshiki Island) where extreme storms passed at the time, partially destroying the barrier beach systems of coastal lakes [148]. Data for Northwestern America indicate that NPH was weakened and/or shifted southward in the late Holocene [25], which seemingly contributed to the shift in the trajectories of most cyclones with sources in the tropical zone of the Western Pacific to the east, without capturing the Southern Russian Far East.

*5.2. LIA*

During the LIA, solar activity decreased significantly (Figure 11a,b) and often changed over time [6,8]. The signals of solar minima are traced in high-resolution proxy records of the Southern Russian Far East (Figure 8). Global cooling was much influenced by volcanic eruptions, several strong volcanic events occurred in the tropics, including a widespread cooling [91,149]. The LIA cooling in the Far Eastern region was close to the cold event 2800–2600 year BP by a temperature decrease (1.5–2 °C below present), but it was longer and unstable (Figures 3, 4 and 6). The coldest subperiod during the LIA corresponds with Bond event 1 (Figure 11). Similar estimates of a decrease in mean annual temperature 2 °C below present were obtained for China [150]. As in Europe [91], the cooling in our region was intensified by the reduction in ocean heat transport to the north, subsequently decreasing the activity of the warm Kuroshio Current [139].

This period in the mainland of the Southern Far East was characterized by high humidity, the river water content increased, and frequent floods occurred (Figures 3 and 5). In lacustrine records of high-temporal resolution, short-term colder episodes stood out when moisture decreased sharply (Figure 8). On Sakhalin, no increase in moisture was recorded, and there were no extreme floods [50]. It was humid on the Kuril Islands, and the amount of winter precipitation increased [111]. On the Shantar Islands, the transition from warm to cold conditions is recorded as ~AD 1310. Moisture increased, as evidenced by higher concentrations of valves and an increase in diatoms living in moister conditions. The proportion of *Sphagnum* mosses increased in swamp vegetation, and dwarf birch was spreading. Humidity decreased slightly ~1450–1690 CE. Findings of marine diatoms in peat signal an increase in storm activity 1310–1380 CE and 1520–1590 CE. In the final phase, it became wetter, the proportion of hydrophilic diatoms and the concentration of valves increased. The climate has become warmer and wetter over the past 210 years [123,151].

The proxy records for East Asia showed that humidification during the LIA was spatially heterogeneous; areas with positive and negative moisture anomalies are distinguished [12,152,153]. In nearby areas, the period was wet. High humidity with frequent floods was observed on the southern and eastern coasts and on the mountains (100 m a.s.l.) of the Korean Peninsula [118,152], Jeju Island [9], and in southern Japan [154]. The mountains in the northern part of the Korean Peninsula and Northeast China (Hani Peat) were moderately dry [12]. In Eastern China, according to chronicle evidence, the wet period lasted from 1240 CE to 1420 CE with a small dry episode 1340–1360 CE [93]. Several drought periods have been identified in the development of Xiaolongwan Lake (655 m a.s.l.) during 1360–1450 CE, 1590–1670 CE, and the last 150 years [155].

The data from the Aleutian Islands, Kamchatka, and Alaska (Figure 11d) show that AL became more intense compared to the MWP and its mode shifted eastward [23,24,136],

although there is no consensus as to the increase in its intensity [23]. According to Nagashima with co-authors [22], AL was strong and shifted to the west. The $\delta^{18}$O records show that the intensity and position of AL were unstable and changed strongly; short-term periods of intensification with a center shift to the east and a weakening with a shift to the west are distinguished [24]. In general, AL was strong but weaker than during the cooling of 2800–2600 year BP [23]. At the eastern position of AL, positive phases of PDO are observed, when the northwestern part of the Pacific became cooler [147]. The trend in the development of a positive PDO phase in the LIA is confirmed by records for the American Northwest [25]. Tree-ring data for the Southern Kurils show that PDO was an important index of large-scale climate variation [109]. The development of such a trend could also intensify cooling and affect the precipitation regime in the Southern Russian Far East.

SH intensified during the LIA, most likely expanded, and shifted eastward. This is evidenced by the activation of aeolian dust transport from the Gobi Desert, which was recorded in the sediments of the Japan Sea [13]. Fine quartz flux from the Taklimakan desert to Jeju Island in Korea decreased sharply [124]. The frequency of dust storms increased in Northern China [156]. Additionally, the deposition flux of aeolian quartz was characterized by sharp changes in time, which indicates the instability of atmospheric circulation and the higher variability of climatic conditions became more variable during the last 700 years. Changes in K+ content of the GISP2 ice-core record from Greenland [4,20] show that the SH of that period was the most intense in the Holocene (Figure 11c). This is also confirmed by $\delta^{18}$O records from lacustrine sediments of Kamchatka [83]. In such a situation, zonal processes of air mass transfer prevailed over the Far East [44,48]. This is evidenced by data for Kamchatka, where cool and wet winters occurred [136] and a notable ice advance was observed in 1350–1850 CE [149].

During the LIA, there were powerful cold outbreaks to the Okhotsk Sea, Northwestern Japan Sea and the adjacent continental regions of the Russian Far East. At present, the extreme cold regime is formed when SH shifts eastward and AL westward, which leads to the formation of a meridional zone of baric gradients, through which cold air actively flows from permafrost regions in Eastern Siberia [157]. The winters of 1999/2000, 2000/2001, 2018/2019, and 2022/2023 CE can be a modern analogue for when cold outbreaks from Yakutia, the cold pole of the Northern Hemisphere, took place. Cold outbreaks recurring during the winter season are characterized by strong northwestern winds and extremely low temperatures [158]. Cold events in the winter of 1999/2000 CE and 2000/2001 CE in Primorye and the northwestern part of the Japan Sea caused a deep convection in the central part of the Japan Sea in February 2000 and the ventilation of the bottom waters of the Japan Sea near the continental slope of Peter the Great Bay at depths of 3000–3200 m in February 2001 [159]. Similar cold outbreaks were observed in the same years in the northwestern part of the Okhotsk Sea. The increase in the frequency of such synoptic situations is due to the SH strengthening, the shift of its center to the east–southeast, as well as the strengthening of the tropospheric cyclone over the Okhotsk Sea [160,161], which can be considered a west–southwest continuation of AL. Such winter synoptic processes could intensify cooling in the LIA.

Thus, in the LIA, two systems intensified; the high pressure gradient between SH and AL caused the activation of the winter monsoon and, most likely, an increase in its duration. A positive correlation between the intensity of the winter monsoon and the SH was noted by many researchers [105]. The 1980s are the modern analogue (even though with a higher temperature background) for when the pressure gradient between AL and SH reached its maximum, and during which the number of "cold" synoptic types of weather increased [162]. Due to such a ratio of strength and position of these centers of atmospheric action, the ice cover of the Okhotsk Sea and Bering Sea increased [135]. Sea ice expansion and a longer period of sea ice development also contributed to the decrease in the temperature regime of the surrounding land. It is possible that, as in Europe [91], winter temperatures were most affected by the decrease. In Northern Japan in the 15–16th centuries, heavy snowfalls were observed according to chronicle data [81].

Heavy snowfalls were also typical for the Kuril Islands [111,115]. It is possible that the intensification of snowfalls is associated with the influence of the Western Pacific Pattern, which must have been predominant in the positive phase during the LIA. This led to an increase in the frequency of snowfalls in Northern Japan [163] and most likely in the Southern Kurils [77,86].

Apparently, heavy snowfalls occurred in Primorye, i.e., deep southern cyclones periodically appeared here. At present, it is typical for the stable location of the SH mode in the east [44]. Heavy snowfalls may be associated with the blocking role of the anticyclone. SH becomes more extensive and powerful when it shifts toward high latitudes [164] and begins to block the exit of polar frontal cyclones from the mainland, while subtropical cyclones from the south and cyclones from the Japan Sea enter the region more often [140].

During more active winter monsoon in the northwestern Pacific, winds intensified, which was recorded in the internal structure of dune fields [165]. Storm activity increase is evidence of strong winds in the past. For example, the salinity rise in Tokotan Lake (Urup Island) ~1335–1450 CE and 1525–1565 CE is associated with an increased storm activity [166]. Strong storm surges were also recorded on the Okhotsk Sea coast of Iturup Island. During strong storms, marine diatoms were brought through the channel to the Lebedinoe Lake [85]. An increase in the amount of allochthonous pollen is also one of the indicators of the strengthening wind regime and active cyclogenesis [123]. Allochthonous arboreal pollen transported from the south was found in peat bogs of the Kuril Islands (Lesser Kurils, Kunashir, Iturup, Ketoi, Shiashkotan, Ekarma, Onekotan islands, ets.) (Figure 7) [84,111]. The transition from the MWP to the LIA was characterized by high storm activity in the Okhotsk Sea, which was restored from the emission of marine aerosols to the Shantar Islands in 1130–1380 CE and 1520–1590 CE. At the beginning and end of the LIA (500–430 CE, 1240–1310 CE, 1450–1520 CE, 1730 CE–second half of the 19th century), cyclones of southwestern, western, and northwestern directions became more active, as evidenced by the supply of allochthonous fir and pine pollen [123].

At present, when SH center occupies an eastern position, in spring and summer, FEL is also shifted to the east and north, and it becomes deeper in spring. The number of cyclones entering the region is usually reduced, but they become more intense. The OH is also becoming more active [44]. Although in general the intensity of the NPH decreased in the late Holocene and/or shifted to the south [25,26], one can assume that during the LIA, the pressure gradient over the Northeast Asian margin and the marginal seas intensified, which led to the activation of the summer monsoon. This may be one of the reasons for the increase in moisture in the summer season. The rise in the summer monsoon intensity is also confirmed by terrestrial data and marine records of South Korea, Eastern China, and southernmost Japan [9,11,12,95,152]. Increase in precipitation is not reflected in some lake records in Honshu Island, Japan [167].

During the LIA, it was humid both on the Kuril Islands and on the mainland, the region was in the zone of active cyclogenesis, and the frequency of strong floods increased. Now in spring, the depth of FEL is a determining factor and has a significant correlation with the intensity and number of cyclones [44]. Apparently, due to the deepening of FEL, the summer extratropical cyclogenesis was also more active. Currently, when OH becomes active, strong floods are observed in Primorye (for example, in 2016 CE), associated with the long-term endurance of deep southern cyclones over the region. On Sakhalin, it was drier during the LIA [51,69] and there were no extreme floods [50]. It is possible that drier conditions in Southern Sakhalin were associated with the blocking effect of OH: high pressure over the Okhotsk Sea became an obstacle to the active exit of cyclones.

In the western part of the Pacific tropical zone, SST (Figure 11f) significantly decreased (1–1.5 °C below the present) [126]; in the area of the Okinawa Trough, a minimum was recorded 1350–1450 CE [125]. At the beginning of the LIA, a recurrence of strong typhoons in Southern Japan increased, which is explained by the activation of El Niño [148]. The El Niño activation phase at the boundary of the MWP and the LIA was also recorded by $\delta^{18}$O in marine sediments near Indonesia [168]. It is possible that dry conditions in

Primorye at the period can be explained by the influence of frequent and intense El Niño (Figure 11g). In the 14th century, the intensity of El Niño decreased sharply [17,27], and ~1320 CE, La Niño and cyclogenesis in the tropical zone of the Pacific became more active, which contributed to the transfer of moist air masses to the continental margin [9].

In Primorye, drier phases associated with colder episodes of the LIA seem to be connected with short periods of El Niño intensification (Figure 11g). Such short periods with more frequent and intense El Niño have been recorded in marine records near Indonesia [168] and in $\delta^{18}$O from corals on the Palmyra Island (a group of Line islands) in the Pacific tropical zone. Moreover, ENSO activity in the middle of the 17th century was not only stronger, but also more frequent compared to the end of the 20th century [169]. Dry phases recorded in high-resolution lacustrine-swamp records of the mountain complexes of Primorye are in good agreement with the frequency of droughts according to the historical documents of the Korean Empire Goryosa (918–1392 CE) and Annals of the Joseon Dynasty (1392–1863 CE) [9].

For the interpretation of our data, records for South Korea are especially important [11,95,119,152] because the typhoons entering there can reach Primorye. The influence of typhoons that emerged in South China (Guangdong province) was less probable [142]. Some dry events (1570–1600 CE, 1700–1800 CE) identified in Primorye [49,170] are close to periods of a decrease in frequency of floods caused by paleotyphoons on the east coast of South Korea (Lagoon Hwajin-po) [118]. The events were likely associated with a decrease in the intensity of the SH [20,153], and hence the intensity of FEL and OH [44].

A wet period identified in the development of the mountain lakes of the Sikhote-Alin (Figure 8) at the beginning of the LIA (1290–1570 CE) most likely corresponds to the contribution of both extratropical cyclones and typhoons coming from the south. A typhoon recurrence graph for the east coast of South Korea shows several periods of floods and heavy rains, as well as low water periods in this age [118]. These dry periods were not expressed in the Sikhote-Alin (Primorye), seemingly due to a high frequency of extratropical cyclones. Wet periods of 1600–1700 CE and in the beginning of the 19th century in the Sikhote-Alin are well correlated with the increase in heavy rainfall frequency in Korea [118]. Grain-size records in the bottom sediments of the East China Sea (the Zhejiang-Fujian mud belt) also show that the intensity of tropical cyclogenesis increased significantly during 1600–1830 CE [120]. It can be assumed that the main contribution to the moisture increase in Primorye was caused by southern cyclones and typhoons. The change in the prevailing tracks of deep cyclones in different periods of the Holocene is a rather complex problem and requires additional high-resolution studies in the region using various proxies.

## 6. Conclusions

The two cold events, close in temperature, caused by solar activity decrease manifested in the Southern Russian Far East with moisture anomalies of opposite signs. Processes in the Pacific interacting with the atmosphere played an important role in the hydroclimatic changes. The difference in paleoclimatic situations can be explained by the proposed patterns. Taking into account numerous paleo-records as well as modern observations in the Pacific region, one can assume that the atmospheric circulation during these cooling periods was different. We believe that the action centers of the atmosphere had different intensities and mode positions. During the cooling 2800–2600 years ago, the pressure gradients between the mainland and the ocean were smaller since the AL and the SH occupied extreme positions (the AL went to the east, and the SH to the west), and in summer, the intensities of the FEL and the NPH were reduced, which could lead to a weakening of the winter and summer monsoons and consequently reduce the amount of precipitation. Cyclones were frequent but not deep. The activation of El Niño, during which abnormally dry years were observed on the mainland, also led to a weakening of cyclogenesis. It is possible that the main trajectories of deep cyclones also shifted eastward, away from the Southern Russian Far East.

The LIA was characterized by strong variability associated both with changes in solar activity and regional factors, as reflected by frequent changes in proxies corresponding to different ecological situations. In general, the pressure gradients between the land and the ocean increased. In winter, the AL intensity grew and the SH was also strong and occupied an eastern position. In the summer, the FEL and the OH became more active. Under these conditions, zonal transfer became more intense. Cooling in the region could have increased due to the positive phase of PDO, as well as a decrease in the intensity of warm currents and a prolonged sea ice cover. Intense snowfalls in the Southern Kurils are possibly related to the positive phase of WPO. The region was in the zone of active cyclogenesis, which was facilitated by the strong variability of interaction in the "ocean-atmosphere" system. Frequent and intense El Niños occurred only at the beginning of the LIA, then La Niño and cyclogenesis in the tropical zone of the Pacific became more active, which contributed to the transfer of moist air masses to the continent margin, including Primorye. The watering of the rivers increased here along with a strong flood frequency. It is possible that extratropical cyclogenesis also became more active. The Kuril Islands were also in the path of deep cyclones and typhoons, unlike Sakhalin Island, where the OH blocked the exit of cyclones.

**Supplementary Materials:** The following supporting information can be downloaded at: https://www.mdpi.com/article/10.3390/cli11040091/s1, Table S1: Proxy records used in this study. Site number as in Figure 2.

**Author Contributions:** Conceptualization and methodology, N.R., L.G. and T.G.; writing of original draft, N.R. and L.G.; diatom analysis and interpretation, T.G.; description of modern climate and atmosphere circulation, V.P.; paleoclimatic interpretation, N.R., L.G. and V.P. All authors have read and agreed to the published version of the manuscript.

**Funding:** This research was funded by the Russian Science Foundation, grant number 22-27-00222.

**Data Availability Statement:** The data are available on request from the authors.

**Acknowledgments:** We are grateful to our colleagues who took part in field work, participated in the processing of the material and reconstructions for key sections. We thank Anna Pshenichnikova (Far East Federal University, Vladivostok, Russia) for assistance with English language editing. The authors are grateful to three anonymous reviewers for providing constructive comments.

**Conflicts of Interest:** The authors declare no conflict of interest.

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
