# Peer review of "“Cold-Dry” and “Cold-Wet” Events in the Late Holocene, Southern Russian Far East"

_climate, doi:10.3390/cli11040091_

Round 1

Reviewer 1 Report

The manuscript carried out a study to understand the effect of hydroclimatic changes specifically the manifestation of cooling during 2800–2600 yr BP and the Little Ice Age (LIA) on diatoms and pollen abundances. The study is extensive using samples from many sites. However, I have some major concerns related to methodology, presentation and interpretations. I regret for my inability to recommend the manuscript for publications in its present condition but encourage authors to clarify/elaborate/address the issues and resubmit. My concerns are given below:

Major concerns

Manuscript has unnecessary and general descriptions throughout. It can be easily reduced to 2/3 rd of its present size.

Presentations in results and discussion are very difficult to read and follow, also difficult to check with the figures. Instead of discussing the findings of individual site, authors may identify the cooling periods and mention what they observed in different sites.

Line 14: It is not clear on the basis of what authors assigned temporal resolution of 26-40 years? How many samples are dated for different sites, what are the materials used for dating, how age increased/changed with depths? A Table with the dates are required (may be as a supplementary). Also provide plots for radiocarbon ages vs depths for different sites.

Line 129-131: some details of the methods particularly how these proxies record the climatic conditions are required.

Line 138-145: what are the implications of these volcanic remains? Please provide details how did you assigned their sources.

The results throughout the manuscript are presented very casually, figures are very busy and difficult to follow.

Presentations of results need improvement, it is difficult to follow the messages authors want to convey.

Minor points

Line 36: please mention the unit of ages (ka BP, BC, AD etc.)

Line 68-104: please separate these paragraphs from introduction and give a separate heading like ‘study area and its present climate’ .

Line 133: please mention what materials was used for radiocarbon dating?

Line 555-616: just general discussion from previous studies, may be removed or included in the introduction. Similarly there are general and irrelevant discussions in many other places.

Author Response

Thank you very much for constructive comments. Our reply is in the file.

Reviewer 2 Report

Dear Authors,

I find your manuscript (MS) presenting important new information on the climatic changes which potentially influence a human civilization. MS combines both broad variety of own data and published archives. It suits scopes of Climate journal.

However, there is a number of concerns, and I would recommend moderate to major revision.

There are many comments within the pdf.

Major concerns:

1. Introduction section must present better 'message' to readers regarding importance of study of Bond events in the region, research approach, and briefly, what is the main 'output'.

On Fig. 1, please, add general atmospheric features (position of High, Low, etc.).

2. Methods section must present more detailed information on the paleoreconstruction routine (indicators, methods...).

3. Results section is too overloaded. It's not easy to follow the text. The chapter must be shorter and more concise. Please, think to reorganize it: give general information on main paleoclimatic indicators which will be described, focus on data on main paleochanges, focus on main areas with well-expressed changes. Sometimes, you give interpretations but they must be in Discussion - no any interpretations in Results!

On Figs 2-3, and 5-6, please, add numbers of lakes, etc. marking on Fig. 1 - this will help readers to see quickly what is what and where.

4. Discussion section overall is very detailed and well-written review of the global/regional paleonvironmental information. Here, I would expect more interpretations of authors' data - mostly they are absent except for subsection 4.2 (but briefly). You could give your interpretations in close correlation to other archives - this will clearly show your own results. Please, give more focus on your own paleostudy.

I would also propose to make a table exhibiting your major findings for the paleoclimate (columns) within intervals of Bond events 2 and 1 (lines) in comparison to global data (separate columns for the mainland and islands).

Author Response

Thank you very much for attention to our paper and constructive comments.

Reviewer 3 Report

Dear authors,

The manuscript “Events “cold-dry” and “cold-wet” in the late Holocene, Southern Russian Far East” present a synthesis based on multi-proxy records of 43 sections, and a paleoclimatic interpretation with proposed changes on atmospheric circulation of two Late Holocene cold events (2800–2600 yr BP and LIA).

I would like to mention how complex it is to carry out paleoclimatic syntheses, and how important it is to have these syntheses in areas with a lack of consistent and reliable paleoclimatic information. Undoubtedly, the contribution of this work will be very valuable for the scientific community, both for those who work in the region and also for contributing to the global understanding of climate dynamics.

The manuscript deserves to be published in Climate. However, I would like to provide some comments that I believe will improve the work and facilitate reading for those readers interested in paleoclimate from other disciplines.

At first, the manuscript is too long (about 4500 words in Results and more than 5000 in Discussion). Both section need to be reduced. The section Results has overlaps with previous work by authors, so that I suggest focusing on the methods used to assemble the database and how to obtain a quality-controlled data. As you pointed out in lines 131-132, usually records have different temporal resolution, chronological control and climate sensitivity. Therefore, it is essential to carry out a quality control of the data as part of the synthesis work. As an example you can check Kaufman et al. 2020 (A global database of Holocene paleo-temperature records. Sci. Data 7, 115) or Ozan et al. 2022 (Disentangling the Medieval Climatic Anomaly in Patagonia and its impact on human societies. Holocene, 32,8: 866-883)

Also, the Figures 2, 3, 5 and 6 are a difficult to read, and different name information is given in the Figures and in the Supplementary Material (e.g. Muta peatbog/ Muta mire). It might be more appropriate to replace the names with numbers in Figures.

It would be interesting to have a figure in the discussion that would help the reader understand the major features of atmospheric changes proposed during the cold events with opposite moisture anomalies.

Author Response

(The authors gave the same response as above.)

Round 2

Reviewer 2 Report

Dear Authors,

to my mind, you made good job correcting the manuscript. It can be accepted.

Author Response

Thank you very much for your attention to our paper and constructive comments.